# Design of Dual-Targeted pH-Sensitive Hybrid Polymer Micelles for Breast Cancer Treatment: Three Birds with One Stone

**DOI:** 10.3390/pharmaceutics15061580

**Published:** 2023-05-24

**Authors:** Degong Yang, Ziqing Li, Yinghui Zhang, Xuejun Chen, Mingyuan Liu, Chunrong Yang

**Affiliations:** 1Department of Pharmacy, Shantou University Medical College, No. 22 Xinling Road, Shantou 515041, China; 2Guangdong Provincial Key Laboratory of Infectious Diseases and Molecular Immunopathology, Shantou University Medical College, No. 22 Xinling Road, Shantou 515041, China; 3Department of Pharmaceutical Sciences, Jiamusi University, 258 Xuefu Road, Jiamusi 154007, China

**Keywords:** dual-targeted, pH-sensitive, hybrid polymer micelles, stability, breast cancer

## Abstract

Breast cancer has a high prevalence in the world and creates a substantial socio-economic impact. Polymer micelles used as nano-sized polymer therapeutics have shown great advantages in treating breast cancer. Here, we aim to develop a dual-targeted pH-sensitive hybrid polymer (HPPF) micelles for improving the stability, controlled-release ability and targeting ability of the breast cancer treatment options. The HPPF micelles were constructed using the hyaluronic acid modified polyhistidine (HA-PHis) and folic acid modified Plannick (PF127-FA), which were characterized via ^1^H NMR. The optimized mixing ratio (HA-PHis:PF127-FA) was 8:2 according to the change of particle size and zeta potential. The stability of HPPF micelles were enhanced with the higher zeta potential and lower critical micelle concentration compared with HA-PHis and PF127-FA. The drug release percents significantly increased from 45% to 90% with the decrease in pH, which illustrated that HPPF micelles were pH-sensitive owing to the protonation of PHis. The cytotoxicity, in vitro cellular uptake and in vivo fluorescence imaging experiments showed that HPPF micelles had the highest targeting ability utilizing FA and HA, compared with HA-PHis and PF127-FA. Thus, this study constructs an innovative nano-scaled drug delivery system, which provides a new strategy for the treatment of breast cancer.

## 1. Introduction

Breast cancer has become the most commonly diagnosed cancer in 2020, and the incidence is rising year after year. According to a report, the numbers of new breast cancer cases and deaths in the United States in 2021 were 284,200 and 43,600, respectively [1]. Breast cancer is mainly caused by malignant changes in the epithelium of breast ducts, which seriously affects the physical and mental health of female patients [2]. At present, there are a variety of treatment methods, such as surgery, radiotherapy, chemotherapy, and molecular targeted therapy. Among them, chemotherapy is an active treatment for all stages of breast cancer, which significantly prolongs the median survival of patients [3]. However, chemotherapeutic drugs along with killing the cancer cells, bring serious damages to the normal cells as well, thereby causing systemic toxicity [4]. Therefore, the development of a novel drug delivery system for targeted and controlled release of chemotherapeutic drugs to tumor sites has attracted widespread attention. 

Nanocarriers are often applied for treating breast cancer [5]. An enzymatically transformable polymer-based nanotherapeutic approach containing colchicine and marimastat is developed to prevent malignant progression of metastatic breast cancer [6]. The exosome membrane coated nanoparticles containing cationic bovine serum albumin conjugated siS100A4 are designed, which significantly inhibits the growth of malignant breast cancer cells [7]. The active tumor targeting nanoparticles containing ferritin and a pH-sensitive molecular is developed to inhibit tumor cell growth and metastasis based on the combination of tumor immunity activation and ferritinophagy-cascade ferroptosis [8].

Polymer micelles are formed by the self-assembly of amphiphilic polymers, which have become one of the most attractive carriers of anticancer drugs [9] because polymeric micelles improve the solubility of insoluble drugs, reduce the toxicity of chemotherapeutic drugs, and improve the stability of drugs in biological media without losing activity [10,11]. However, maintaining the integrity of the polymer micelles in the circulation and for the drug release at the action site remain challenging [12]. The bloodstream causes the dilution of polymer micelles, thereby facilitating the premature release of the payload in the bloodstream [13]. When the temperature of the system is elevated above the glass transition temperature of the polymer micelles, the critical micelle concentration value is increased, which contributes to a liquid-like state of the micellar core and reduces stability [14]. Therefore, it is particularly important to design a safe and efficient polymer micelle for enhancing the stability, controlled release ability and targeting ability. A previous study showed that changing the surface charge of micelles with isomaltodextrin can improve stability [15]. Another study constructed the pH-sensitive polymeric micelles assembled for drug delivery by stereo complexation between PLLA-b-PLys and PDLA-b-mPEG [16]. To improve the targeting of micelles, yet another study devised a targeted polyelectrolyte complex micelle to deliver therapeutic nucleotides to inflamed endothelium in vitro by displaying the peptide VHPKQHR targeting VCAM-1 [17]. The obtained results highlight the urgent need to scientifically design efficient polymeric micelles for achieving the aforementioned three goals, that is, improved stability, controlled release ability and targeting ability. 

This study aims to develop the dual-targeted pH-sensitive hybrid polymer micelles for improving the stability, controlled-release ability and targeting ability. A previous study demonstrated that hybrid micelles formed by mixing of two or more kinds of polymers exhibited higher stability than single-component polymer micelle [18,19]. Additionally, polyhistidine (PHis) was designed and synthesized for pH-sensitive controlled release [20]. In contrast, this study uses folic acid (FA) and hyaluronic acid (HA) to construct the dual-targeted polymer micelle. In a previous study, magnetic carbon nanospheres modified by FA were developed for the targeted delivery of adriamycin [21]. Sun constructed an HA-targeting drug delivery system based on a metal-organic skeleton for efficient antitumor therapy [22]. Taking these into account, this study prepared the mixed polymer micelles (HPPF) via HA-PHis and Plannick-FA (PF127-FA), which were shown in Figure 1. The anticancer activity of docetaxel (DTX) was five times higher than that of paclitaxel, but its water solubility was still poor, which did not achieve the concentration requirements of clinical application [23]. Therefore, it is expected to increase the solubility of DTX utilizing hybrid polymer micelles.

## 2. Materials and Methods

### 2.1. Materials, Cell Lines, and Animals

#### 2.1.1. Materials

HA (*M*w = 10,000), thionyl chloride, tetrahydrofuran, isopropylamine, N, N-Dimethylformamide (DMF), N, N′-Carbonyldiimidazole (CDI), ethylenediamine and FA were obtained from Macklin Biochemical Co., Ltd. (Shanghai, China). N_α_-CBZ-Nim-DNP-L-histidine, 1-Ethyl-3-(3-dimethylaminopropyl) carbodiimide (EDC), N-Hydroxysuccinimide (NHS) and N, N′-Dicyclohexylcarbodiimide (DCC) were obtained from GL Biochem Ltd. (Shanghai, China). PF127 was obtained from the BASF (Shanghai, China). DTX was obtained from the Jinhe Biotechnology Co., Ltd. (Shanghai, China). 3-(4,5-dimethyl-2-thiazolyl)-2,5-diphenyl tetrazolium bromide (MTT), coumarin 6 (Cou-6) and 4′,6-diamidino-2-phenylindole (DAPI) were obtained from Sigma (St. Louis, MO, USA).

#### 2.1.2. Cell Lines and Animals

HepG2 (human liver cancer cells) and MCF-7 (human breast cancer cells) were purchased from the Cell Bank of the Chinese Academy of Sciences (Shanghai, China). All cells were cultured in DMEM medium (Gibco, Thermal Fisher, Lenexa, TX, USA) supplemented with 8% fetal bovine serum (Gibco, Thermal Fisher, Lenexa, TX, USA) and 1% penicillin-streptomycin in a humidified atmosphere of 95% air and 5% CO_2_ at 37 °C, respectively. 

Female BALB/c mice (18 ± 2 g) were purchased from the laboratory animal center of Shantou University Medical College (Shantou, China). All operational processes were carried out according to the NIH Guidelines for the Care and Use of Laboratory Animals and were approved by the Animal Ethics Committee of Shantou University Medical College (SUMC2022-152).

### 2.2. Synthesis and Characterization of HA-PHis

#### 2.2.1. Synthesis and Characterization of Nim-DNP-L-Histidine

Briefly, Nα-CBZ-Nim-DNP-L-histidine was dissolved in anhydrous tetrahydrofuran, then thionyl chloride was added to react for 5 h. Finally, the products were obtained by filtration and recrystallization, and the structure was characterized using ^1^H NMR.

#### 2.2.2. Synthesis and Characterization of Poly (Nim-DNP-L-Histidine)

Nim-DNP-L-histidine was dissolved in DMF containing isopropylamine, and the solution was reacted under N_2_ at room temperature for 4 days. Next, the solution precipitated in the cold diethyl ether. Finally, the poly (Nim-DNP-L-histidine) was obtained by solvent evaporation and characterized using ^1^H NMR.

#### 2.2.3. Synthesis and Characterization of HA-PHis

HA was dissolved in anhydrous formamide at 55 °C, then cooled to room temperature, then NHS and EDC were added to react for 2 h on ice. Subsequently, poly (Nim-DNP-L-histidine) was dissolved in DMF and added to HA solution to react for 48 h at room temperature. The mixture was dialyzed with distilled water for 3 days and lyophilized under vacuum. Next, the mixture was dissolved in anhydrous formamide containing mercaptoethanol to react for 48 h at room temperature for removing 2, 4-dinitrophenyl from poly (Nim-DNP-L-histidine). Finally, the HA-PHis were dialyzed with distilled water for 3 days and lyophilized. The structure of HA-PHis was characterized using ^1^H NMR.

### 2.3. Synthesis and Characterization of PF127-FA

#### 2.3.1. Synthesis and Characterization of CDI-PF127 

An appropriate amount of PF127 was dissolved in acetone and precipitated by precooled n-hexane. The purified PF127 was obtained by vacuum drying, then dissolved in anhydrous acetonitrile. In addition, the CDI was dissolved in anhydrous acetonitrile, then slowly dripped into PF127 anhydrous acetonitrile solution within 2 h under nitrogen, for 4 h. Afterwards, it was concentrated by rotary evaporation and washed three times with precooled ether. The CDI-PF127 was collected by vacuum drying, and characterized using ^1^H NMR.

#### 2.3.2. Synthesis and Characterization of NH_2_-PF127

CDI-PF127 was dissolved in anhydrous acetonitrile. The ethylenediamine was slowly dripped into the above solution within 3 h and stirred overnight at room temperature. The excess ethylenediamine was removed by rotary evaporation and washed with precooled ether three times. The white crystalline powder (NH_2_-PF127) was obtained by vacuum drying, and characterized using ^1^H NMR.

#### 2.3.3. Synthesis and Characterization of PF127-FA

NH_2_-PF127 was dissolved in anhydrous DMSO, then added to triethylamine as the liquid A. FA, NHS and DCC were dissolved in DMSO, and triethylamine was added and reacted for 10 h under magnetic stirring at room temperature (liquid B). Liquid B was slowly added to liquid A under the protection of nitrogen and stirred overnight at room temperature. The deionized water was slowly dripped into the reaction solution to remove the unreacted FA. The supernatant was dialyzed with deionized water for 3 days. The yellowish solid powder (PF127-FA) was obtained by freeze-drying, and characterized using ^1^H NMR.

### 2.4. Preparation and Characterization of Micelles

HPPF micelles were prepared using the film dispersion method [24]. The copolymers were dissolved in acetonitrile, then dried. The mixing ratios of HA-PHis and PF127-FA were shown in Table 1. The optimized prescription of HPPF micelles was determined according to particle size and zeta potential. The particle size and zeta potential of HPPF micelles were determined via the Malvern particle size analyzer (Malvern, UK). The morphology of micelles was observed using transmission electron microscope (TEM). 

The entrapment efficiency (*EE*%) and drug loading (*DL*%) of HPPF micelles were determined according to Formulas (1) and (2).
(1)$$EE\%=(1-\frac{C_{free}}{C_{total}})\times 100\%$$
(2)$$DL\%=\frac{W_{drug}}{W_{lipid}}\times 100\%$$
where, *C*_free_ was the concentration of free DTX (μg/mL); *C*_total_ was the total concentration of DTX in the suspension (μg/mL); *W*_drug_ was the amount of drugs encapsulated in HPPF micelles (mg); and *W*_lipid_ was the weight of mixed carrier material in the prescription (mg).

Pyrene was used to determine the critical micelle concentration (CMC) of HPPF micelles. When the polymer concentration was greater than a certain value, the excitation wavelength shifted from 334 nm to 336 nm. The different volumes of polymer solution were added to the pyrene, and the polymer concentration range was 10^−4^−10^−1^ g/L. 

### 2.5. In Vitro Drug Release 

The drug-loaded micelles were added into the dialysis bag (interception of molecular weight: 12,000 Da), then placed in the PBS release medium. The medium was removed and the equal amount of fresh-release medium was replenished. The drug content in the release medium was determined via HPLC, and the cumulative release percent was calculated according to the Formula (3).
(3)$$E_{r}=\frac{V_{e}\sum_{i-1}^{n-1}C_{i}+V_{0}C_{n}}{m_{drug}}$$
where, *E_r_* was cumulative drug release amount (%); *V_e_* was replacement volume of PBS (mL); *V*_0_ was total volume of release medium (mL); *C_i_* was concentration of release solution during the *i* h displacement sampling (μg/mL); *m*_drug_ was total mass of drugs carried (mg); and *n* was number of replacement PBS.

### 2.6. Cytotoxicity 

HepG2 and MCF-7 cells were chosen to evaluate cell cytotoxicity of blank HPPF and HPPF/DTX. The cell inoculation density was 6 × 10^4^ cells·mL^−1^, and the blank control group was the serum-containing medium group. The blank HA-PHis, blank PF127-FA, and blank HPPF (8:2) were added, and the concentration was 80, 40, 20, 10, and 5 μg·mL^−1^, respectively. The HA-PHis/DTX, PF127-FA/DTX, and HPPF/DTX were added, and the concentration was 20, 15, 10, 5, 2, and 1 μg·mL^−1^, respectively. The absorbance of wavelength 492 nm was determined using the enzyme-labeling instrument. The cell survival rate was calculated according to the Formula (4).
(4)$$Cell\;survival\;rate\%=\frac{{OD}_{experimenta\;group}}{{OD}_{control\;group}}$$
where, *OD* was optical density. 

### 2.7. In Vitro Cellular Uptake

HepG2 and MCF-7 cells in the logarithmic phase were inoculated at a concentration of 1 × 10^5^ cells·mL^−1^. Next, 100 μg·mL^−1^ of HA-PHis, PF127-FA, and HPPF containing coumarin-6 were added. DAPI was added for nucleus staining, and the cell uptake was observed using the laser confocal microscope. 

### 2.8. In Vivo Fluorescence Imaging and Tissue Distribution

MCF-7/ADR tumor-bearing mice were injected with 200 μL HA-PHis, HPPF, and Dir fluorescence markers, respectively. At 0.5, 6, 12, 24, and 48 h, the fluorescence intensity of tumor site in mice was monitored using fluorescence imaging. After 48 h, the mice were killed and main organs (heart, liver, spleen, lung, kidney, and tumor) were washed with normal saline three times. Then, the fluorescence intensity of organ was measured.

### 2.9. Statistical Analysis 

Results were expressed as mean ± S.D. The data were subjected to analysis of variance (ANOVA) using SPSS 21.0 software. *p* < 0.05 was taken as a significant level.

## 3. Results and Discussion

### 3.1. Characterization of HA-PHis and PF127-FA

According to the characteristic peaks in Figure 2a, *δ*_D_ = 9.18 ppm (-N=CH), *δ*_E_ = 7.71 ppm (-N-CH=C-), *δ*_G_ = 4.83 ppm (-CH-), and *δ*_F_ = 3.20 ppm (-CH_2_-); thus indicating that NCA was synthesized successfully [25]. By comparing Figure 2a,b the appearance of isopropyl characteristic peak (*δ*_H_ = 0.79 ppm) indicated that PHis-DNP was formed [26]. The characteristic peaks in Figure 2c, *δ*_C_ 2.02 ppm (-COCH_3_-), *δ*_D_ 1.30–1.39 ppm (-C-CH_3_-), *δ*_B_ 7.34 ppm (-N-CH=C-) and *δ*_A_ 8.64 ppm (-N=CH-) indicated that HA-PHis was successfully synthesized [27]. 

The ^1^H-NMR spectrum of PF127, FA, physical mixture of PF127 and FA, and PF127-FA were shown in Figure 3. The characteristic peaks of PF127 were *δ*_A_ 3.38 ppm (CH_2_CH(CH_3_)O), and *δ*_B_ 3.51 ppm (CH_2_CH(CH_3_)O). The characteristic peak shift of FA was *δ*_A_ 11.48 ppm (OH) [28]. The characteristic peak shift of physical mixture of PF127 and FA was 11.61 ppm, which illustrated that FA was covalently bound to PF127 [28]. In addition, the characteristic peak shift of OH (FA) disappeared, which proved that the COOH of FA interacted with the PF127 through the covalent bond. It proved that PF127-FA was synthesized. 

### 3.2. Characterization of Micelles

#### 3.2.1. Particle Size and Zeta Potential 

The HPPF micelles were prepared using HA-PHis and PF127-FA, and particle size varied with the mass ratio of two block copolymers (Table 2). When the mass ratio was 5:5 and 6:4, two block polymers existed separately as single-component micelles. It demonstrated that they were not well assembled into hybrid polymer micelles [29]. When the mass ratio was 8:2 and 9:1, the hybrid polymer micelles with uniform particle size and good dispersion were formed. When the mass ratio was 9:1, the value of zeta potential was lower than that of 8:2. When the absolute value of zeta potential was higher, the electrostatic repulsive force between the particles was greater [30]. Therefore, the mixed micelles with a mass ratio of 8:2 was selected as the optimized prescription for next studies (PDI: 0.19 ± 0.06). In addition, the stability of HPPF (−17.4 ± 0.9 mV) was significantly enhanced compared with HA-PHis (−13.2 ± 7.8 mV) and PF127-FA (−8.5 ± 1.1 mV) (*p* < 0.05), which proved that the strategy using hybrid polymer micelles was successful.

#### 3.2.2. Morphological Observation 

The shape of HPPF/DTX and HA-PHis was observed using TEM and shown in Figure 4. The shape of HPPF/DTX was spherical and the distribution was uniform. The particle size of HPPF micelles was slightly larger than that of HA-PHis micelle. The reason was that PF127-FA and HA-PHis were self-assembled into HPPF micelles in an embedded form. The hydrophilic chain of HA-PHis was exposed owing to the long chain of PF127-FA, which caused the larger particle size [31]. 

#### 3.2.3. Entrapment Efficiency and Drug Loading 

The entrapment efficiency and drug loading of HPPF micelles were 87.2 ± 1.9% and 6.0 ± 0.1%, respectively, which were higher than HA-PHis (84.8 ± 2.1% and 4.2 ± 0.1%). The PF127-FA increased the proportion of hydrophobic blocks of the micelle core, which was beneficial to the loading of hydrophobic drugs (DTX). This study showed that the length of the hydrophobic blocks was closely related to drug loading [32]. Thus, HPPF micelles improved the poor solubility of DTX.

#### 3.2.4. Determination of Critical Micelle Concentration 

The aggregation behavior of HPPF micelles was investigated by measuring the fluorescence spectral curve of pyrene (Figure 5a). The critical micelle concentration of HPPF micelles was 0.04 mg·mL^−1^. The lower critical micelle concentration was beneficial for the stability of micelles in vivo [33]. This study showed that the CMC value of micelles was an important factor that signified the stability, and that a lower CMC value provided greater solubilization of loaded payload [34].

#### 3.2.5. In Vitro Drug Release 

The in vitro drug release experiments were performed to investigate the pH-sensitive release of HPPF micelles in phosphate buffers with different pH values (7.4 and 5.0). As expected, more than 90% of the free drugs were released from DTX solution within 8 h at pH 7.4 (Figure 5b). However, within 72 h, only 45% of DTX was released from the HPPF micelles, which indicated that HPPF micelles ensured long-term stability in the bloodstream and prolonged the circulation time [35]. At pH 5.0, nearly 90% of DTX was liberated from the HPPF micelles within 8 h of incubation, which was in good agreement with previous studies [36]. The p*K*a value of histidine was close to the tumor site acidic environment, which caused protonation and soluble transformation [37]. Hence, the pH sensitivity of PHis in HPPF micelles was confirmed. In addition, in vitro drug release behaviors were all consistent with the Higuchi model (*r* = 0.9545, *r* = 0.9573, and *r* = 0.9521), indicating that drugs were released through diffusion from the micelles [38]. In summary, the experiments proved the pH-sensitive behavior of the HPPF micelles.

### 3.3. Cytotoxicity 

The effects of blank HA-PHis, PF127-FA and HPPF on the growth of HepG2 and MCF-7 cells were determined via MTT (Figure 6a,b). With the increase in the concentration, the survival rates of HepG2 and MCF-7 cells did not change significantly (*p* > 0.05), indicating that blank HA-PHis, PF127-FA and HPPF had no obvious cytotoxic effect on HepG2 and MCF-7 cells.

Then, effects of micelles containing DTX on the cell survival rate in HepG2 and MCF-7 cells were evaluated and results were shown in Figure 6c,d. It was found that toxic effects were dependent on the concentration of micelles. For the HepG2 cells, the cytotoxicity of the micelles was ranked as follows: HPPF/DTX (IC_50_: 1.7 μg/mL) > PF127-FA/DTX (IC_50_: 2.5 μg/mL) > HA-PHis/DTX (IC_50_: 4.6 μg/mL). This is because FA was specifically targeted on the surface of tumor cells, and FA receptor was highly expressed on the surface of tumor cells [39,40]. However, HA-PHis had no targeting ability to HepG2 cells owing to low expression of the CD44 receptor [41]. For the MCF-7 cells, the cytotoxicity of the micelles was ranked as follows: HPPF/DTX (IC_50_: 4.2 μg/mL) > HA-PHis/DTX (IC_50_: 7.7 μg/mL) > PF127-FA/DTX (IC_50_: 10.3 μg/mL). HA-PHis had targeting ability to MCF-7 cells, because the CD44 receptor were overexpressed on the surface of the MCF-7 tumor [42]. Hence, the HPPF owned the highest targeting ability utilizing the FA and HA, which formed more DTX and killed tumor cells. 

### 3.4. In Vitro Cellular Uptake 

Cellular uptake of HPPF micelles were observed via laser confocal localization using HepG2 and MCF-7 cells. Coumarin-6 carrier was chosen as the probe. After incubation for 2 h, the fluorescence intensity of HepG2 cells was very dark, and the HPPF micelles were distributed in the cytoplasm of the cells, but not in the nucleus (Figure 7a). It was suggested that HPPF was swallowed into the cytoplasm by cells, but the uptake was very small [43]. The fluorescence intensity of MCF-7 cells was much stronger than that of HepG2 cells (Figure 7b). The HPPF micelles was mainly distributed in the cytoplasm, but not in the nucleus. The results showed that HPPF micelles were effectively swallowed endocytosis into the cytoplasm by MCF-7 cells. Additionally, the uptake was significantly higher than that of HepG2, which was in good agreement with the results of cytotoxicity. It also proved that only FA did not ensure that the prepared micelles were targeted to the tumor site. The previous study also proved that the conjugation of mesoporous silica nanoparticles with FA increased the efficiency of nanoparticles entering the cell and localization in the close vicinity of the nucleus [44]. The results of confocal microscopy proved that the HA-receptor mediated cellular uptake of redox-sensitive chitosan-based nanoparticle [45].

### 3.5. In Vivo Fluorescence Imaging and Tissue Distribution 

The targeting ability of HPPF micelles to tumors in mice was observed via in vivo fluorescence imaging. The HPPF micelles were distributed all over the body after injection for 0.5 h (Figure 7c). With the extension of time, the Dir fluorescence were transferred to the liver, spleen, and other organs. Additionally, the enhancement of fluorescence intensity indicated the accumulation of HPPF. After 12 h, the fluorescence intensity of tumor reached the peak, which was significantly higher than other tissues and organs. In addition, the HPPF fluorescence intensity of the tumor site was significantly higher than that in other organs, indicating good tumor targeting (Figure 7d). Compared with HPPF, the fluorescence intensity of HA-PHis was significantly weakened, indicating that HPPF increased the drug accumulation in tumor site and prolonged the accumulation time of drug in the tumor site. This is mainly attributed to the dual-targeted action of HA and FA, which effectively solved the off-target phenomenon. For example, the study developed high-efficiency dual-targeted nanoflowers containing ferroferric oxide and HA, which improved the specific uptake of drugs at tumor site by the dual action of CD44 ligand HA and magnetic nanoparticles guided by magnetic force [46].

## 4. Conclusions

In this study, a novel dual-target pH-sensitive HPPF hybrid micelle was successfully constructed. The optimal mixing ratio (HA-PHis: PF127-FA = 8:2) was obtained according to particle size and zeta potential. The HPPF micelles improved the stability with higher zeta potential and lower critical micelle concentration. The pH-sensitive release of HPPF micelles was demonstrated owing to histidine protonation. In vivo image demonstrated that the targeting ability of HPPF micelles was higher than FA and HA. In conclusion, this study provided a new strategy for the development of polymer micelle, which reduced the side effects of chemotherapeutic drugs and improved the treatment of breast cancer.

## Figures and Tables

**Figure 1 pharmaceutics-15-01580-f001:**
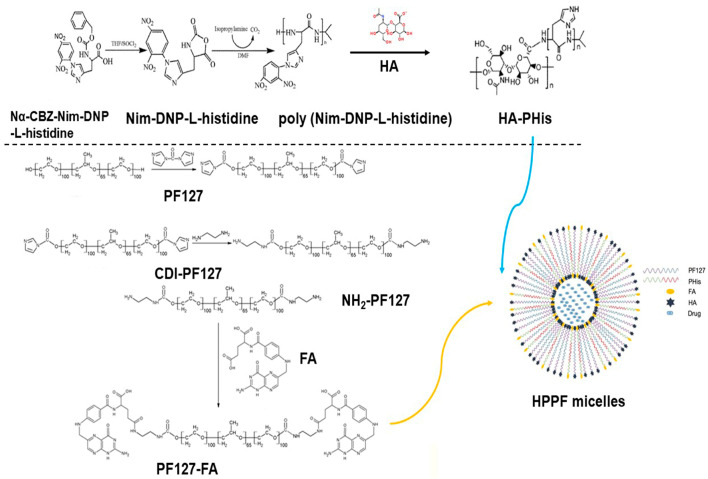
The structures of the mixed polymer micelles.

**Figure 2 pharmaceutics-15-01580-f002:**
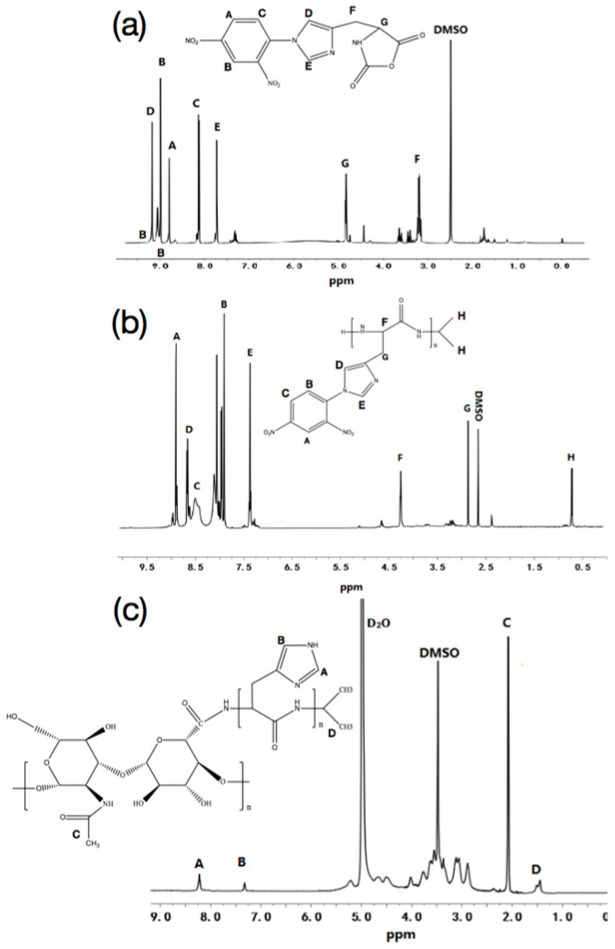
(**a**) ^1^H-NMR spectrum of Nim-DNP-L-histidine, (**b**) ^1^H-NMR spectrum of poly (Nim-DNP-L-histidine), (**c**) ^1^H-NMR spectrum of HA-PHis.

**Figure 3 pharmaceutics-15-01580-f003:**
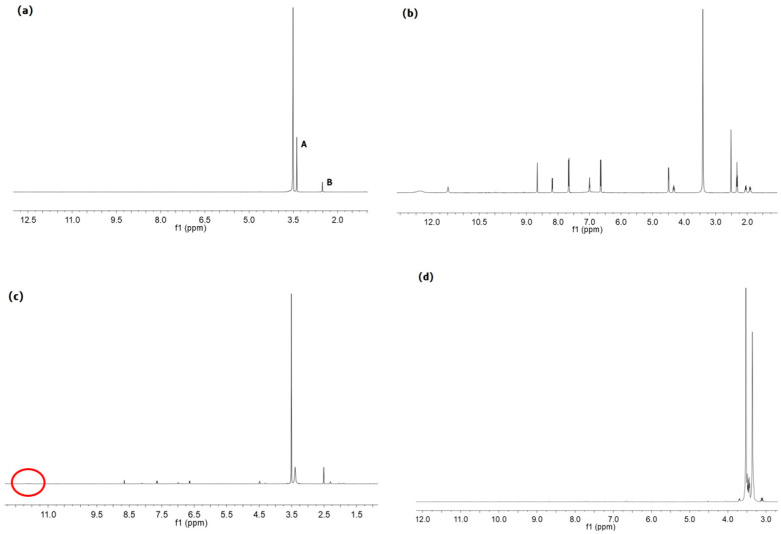
(**a**) ^1^H-NMR spectrum of PF127, (**b**) ^1^H-NMR spectrum of FA, (**c**) ^1^H-NMR spectrum of physical mixture of PF127 and FA, (**d**) ^1^H-NMR spectrum of PF127-FA.

**Figure 4 pharmaceutics-15-01580-f004:**
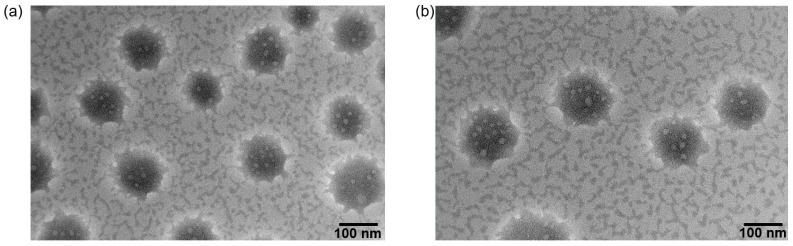
Morphological observation of HA-PHis (**a**) and HPPF/DTX (**b**) via TEM imaging.

**Figure 5 pharmaceutics-15-01580-f005:**
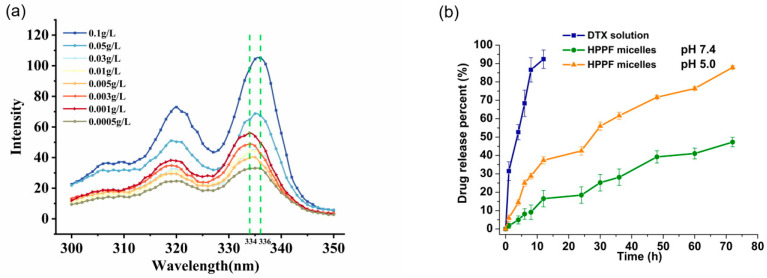
(**a**) The critical micelle concentration of HPPF micelles and (**b**) the drug release profiles from micelle.

**Figure 6 pharmaceutics-15-01580-f006:**
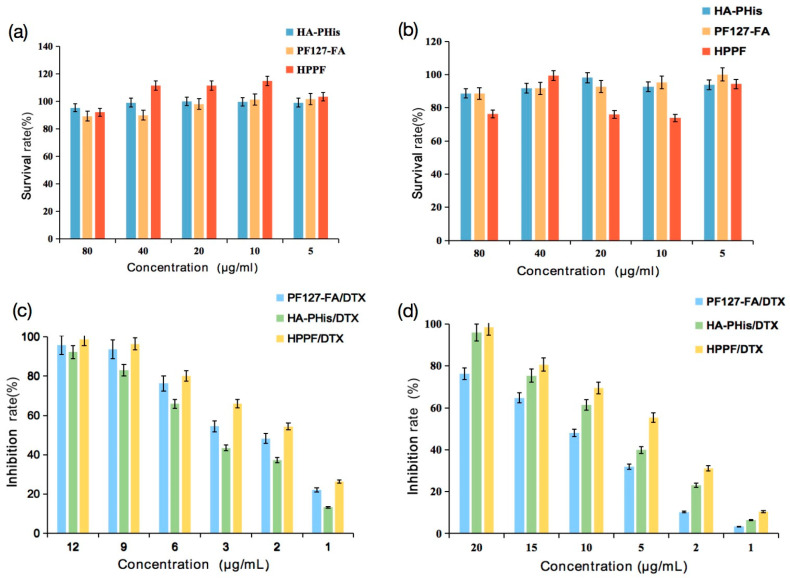
(**a**,**b**) Effects of concentration of blank carriers on HepG2 and MCF-7 cells survival rate (*n* = 5), (**c**,**d**) Effects of concentration of carriers containing DTX on cell survival rate in HepG2 and MCF-7 cells (*n* = 5).

**Figure 7 pharmaceutics-15-01580-f007:**
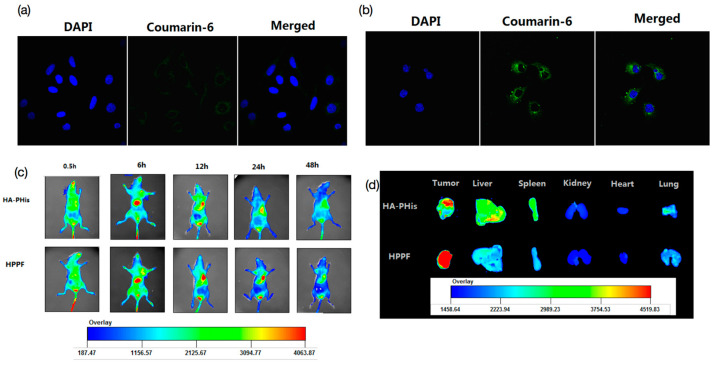
Cellular uptake of HPPF micelles using HepG2 (**a**) and MCF-7 cells (**b**), in vivo imaging results of HPPF micelles (**c**), the ex vivo images of organs (**d**).

**Table 1 pharmaceutics-15-01580-t001:** The mixing ratios of HA-PHis and PF127-FA.

Sample	HA-PHis:PF127-FA
A	9:1
B	8:2
C	7:3
D	6:4
E	5:5

**Table 2 pharmaceutics-15-01580-t002:** Particle size and zeta potential of polymer micelles (*n* = 3).

Sample	Particle Size (nm)	Zeta Potential (mV)	Single/Double Peak
HA-PHis	119.1 ± 7.7	−13.2 ± 7.8	Single peak
PF127-FA	40.0 ± 1.2	−8.5 ± 1.1	Single peak
A (9:1)	115.3 ± 6.1	−16.2 ± 0.8	Single peak
B (8:2)	119.6 ± 6.3	−17.4 ± 0.9	Single peak
C (7:3)	143.0 ± 6.3	−17.1 ± 3.2	Irregular single peak
D (6:4)	131.7 ± 7.0	−21.0 ± 2.5	Double peak
E (5:5)	122.3 ± 8.4	−19.2 ± 3.1	Double peak

## Data Availability

The data is unavailable due to privacy or ethical restrictions.

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
