# Peer review of "Design of Dual-Targeted pH-Sensitive Hybrid Polymer Micelles for Breast Cancer Treatment: Three Birds with One Stone"

_pharmaceutics, 2023, doi:10.3390/pharmaceutics15061580_

Round 1

Reviewer 1 Report

1. A major issue of this study is quality of presentation, particularly for introduction. The authors should introduce more background on treatment of breast cancer by using nanocarrier (10.1002/adma.202105254). Current introduction is too general and  is quite jargon heavy. 

2. It is better to emphasize that the reported delivery system can achieve multiple functions (10.1021/jacs.0c09029). Current discussion is not attractive.

3. PDI of DLS results should be given.

4. In Section 2.6 and Figure 5, please indicate the concentration of which substance.

5. For cytotoxicity, please calculate IC50.

6. The authors should include proper controls to prove double ligand effect when doing cytotoxicity and cellular uptake experiments.

7. Please provide quantitative results for cellular uptake and biodistribution.

8. If possible, please provide antitumor results.

9. Breast cancer has become the most commonly diagnosed cancer in 2020. The authors can introduce this.

Author Response

The reply is uploaded.

Reviewer 2 Report

The Manuscript by D. Yang, Z. Li, Y. Zhang, X. Chen, M. Liu, C. Yang “Design of dual-targeted pH-sensitive hybrid polymer micelles for breast cancer treatment: Three birds with one stone” describes the design of innovative nano-scaled drug delivery system allowing selective targeting breast cancer cells. Development of novel highly-selective tumor-targeting agents, including micelles and nano-carriers represents an actual task. In the Manuscript, an elegant promising approach to anti-tumor drug delivery to HepG2 an MCF-7 cells is described. Micelles were characterized (morphology, zeta-potential, drug pH-dependent release profile) Both in vitro and in vivo experiments clearly demonstrates the advantages of the use of hybrid polymer micelles as a drug delivery system.

However, the characterization of chemical building blocks for micelles is rather poor. NMR spectra are not described in detail and hardly visible in Manuscript (Fig. 2).

The structure of compound PF127-FA rises some doubts. Of course, it is hard to prove that folic acid is covalently bound to PF127. NMR spectra of covalently bound conjugates are likely to be similar to those of a simple mixture of reactants; activation of carboxyclic groups of folic acid with DCC/NHS can also afford some side-compounds. At least NMR spectra of folic acid and/or its salt should be used as references. So, the dependence of cytotoxicity of micelles on a relative content of folic acid receptors on the cells’ surface as well as the statement “Only FA did not ensure that the prepared micelles were targeted to the tumor site” can not be related with folic acid residues content in hybrid micelles. It remains unclear, whether folic acid residues covalently bound to PF127 and are inseparable from micelles OR folic acid (or related compound) was just adsorbed on the surface of particles. More strong evidences are needed to prove the structure of folic acid conjugate.

Moreover, some comparison with biodistribution of known folic acid and hyaluronic acid derivatives should probably be done. In Results and Discussion, authors can discuss some known examples of intracellular distribution of folic acid or hyaluronic acid containing nano-carriers. It is done at some extent, but mentioning of several more examples would be illustrative.

I believe that the Article from is of interest to a broad auditory of the Pharmaceutics journal, but it can not be published in its present form. However, it deserves publishing after considering the comments above.

Author Response

The reply is uploaded.

Round 2

Reviewer 1 Report

Labels of name is 'a' 'b' 'c'. But labels of affiliation is '1' '2' '3'.

Reviewer 2 Report

The manuscript can be published in its present form.